# Lung Ultrasound in Neonatal Respiratory Distress Syndrome: A Narrative Review of the Last 10 Years

**DOI:** 10.3390/diagnostics14242793

**Published:** 2024-12-12

**Authors:** Federico Costa, Annachiara Titolo, Mandy Ferrocino, Eleonora Biagi, Valentina Dell’Orto, Serafina Perrone, Susanna Esposito

**Affiliations:** 1Pediatric Clinic, Parma University Hospital, Department of Medicine and Surgery, University of Parma, 43126 Parma, Italy; costa.fede95@gmail.com (F.C.); titolo.annachiara@gmail.com (A.T.); mandy.ferrocino@unipr.it (M.F.); eleonora.biagi@unipr.it (E.B.); 2Neonatology Unit, Parma University Hospital, Department of Medicine and Surgery, University of Parma, 43126 Parma, Italy; valentinagiovanna.dellorto@unipr.it (V.D.); serafina.perrone@unipr.it (S.P.)

**Keywords:** lung ultrasound, neonates, lung ultrasound score, respiratory distress syndrome, neonates, surfactant therapy

## Abstract

Neonatal respiratory distress syndrome (RDS) is a common and potentially life-threatening condition in preterm infants, primarily due to surfactant deficiency. Early and accurate diagnosis is critical to guide timely interventions such as surfactant administration and respiratory support. Traditionally, chest X-rays have been used for diagnosis, but lung ultrasound (LUS) has gained prominence due to its non-invasive, radiation-free, and bedside applicability. Compared to chest X-rays and CT scans, LUS demonstrates superior sensitivity and specificity in diagnosing RDS, particularly in identifying surfactant need and predicting CPAP failure. Additionally, LUS offers real-time imaging without radiation exposure, an advantage over other modalities. However, its broader adoption is limited by challenges in standardizing training, ensuring diagnostic reproducibility, and validating scoring systems, especially in resource-limited settings. This narrative review aims to evaluate the role of LUS in the diagnosis and management of neonatal RDS over the past decade, focusing on its clinical utility, scoring systems, and emerging applications. We reviewed the literature from 2013 to 2023, focusing on studies evaluating LUS’ diagnostic accuracy, scoring systems, and its potential role in guiding surfactant therapy and predicting CPAP failure. Despite its benefits, addressing the variability in operator expertise and integrating artificial intelligence to enhance usability are crucial for ensuring LUS’ efficacy across diverse clinical environments. Future research should prioritize standardizing training and scoring protocols to facilitate wider implementation and optimize neonatal respiratory care outcomes.

## 1. Introduction

Neonatal respiratory distress syndrome (RDS) is a prevalent and potentially life-threatening condition that primarily affects preterm infants but can also occur in full-term newborns. RDS is characterized by increasing respiratory difficulty, typically beginning at or shortly after birth, and often intensifies within the first 48 to 72 h of life before gradually improving. The incidence and severity of RDS are inversely correlated with gestational age; the more premature the infant, the higher the risk and severity of the condition. However, RDS can still manifest in full-term neonates due to various perinatal complications [1,2].

The primary pathophysiological cause of RDS is a deficiency of pulmonary surfactant, a substance critical for reducing alveolar surface tension and preventing lung collapse. Without adequate surfactant, neonates are prone to the development of diffuse atelectasis and the formation of eosinophilic hyaline membranes, which impair gas exchange and lead to progressive respiratory failure. Clinically, RDS presents with signs of respiratory distress such as cyanosis, tachypnea, nasal flaring, grunting, and chest retractions. Complications such as pulmonary edema, which can occur due to capillary leakage and fluid retention, may further worsen respiratory function [3].

The standard treatment for RDS includes the antenatal administration of corticosteroids, the early initiation of continuous positive airway pressure (CPAP), and exogenous surfactant replacement to restore alveolar stability. Advances in neonatal care have significantly reduced the need for invasive mechanical ventilation, though it remains necessary in severe cases or when complications such as persistent pulmonary hypertension (PPHN), pneumothorax, or pulmonary hemorrhage arise. However, invasive ventilation is associated with increased risks, particularly bronchopulmonary dysplasia (BPD) [4].

While chest X-ray (CXR) has long been the traditional imaging method for diagnosing RDS, its limitations—such as radiation exposure and lack of specificity in some cases—have prompted interest in alternative diagnostic tools. Lung ultrasound (LUS) has emerged as a promising modality for evaluating neonatal lung conditions, offering numerous advantages including real-time imaging, radiation-free diagnostics, and non-invasiveness [5]. Importantly, LUS provides high diagnostic accuracy for conditions like pneumothorax, pleural effusion, and RDS [6]. It can visualize key pathological features, such as lung consolidation and atelectasis, and has shown superior performance compared to CXR in several studies [7].

Recent research has focused on standardizing LUS protocols and scoring systems, such as the lung ultrasound score (LUSco), to reduce subjectivity and improve diagnostic reproducibility [8,9]. Moreover, LUS has demonstrated a potential role in guiding therapeutic decisions, such as the timing of exogenous surfactant administration, and in predicting the failure of non-invasive respiratory support in neonates [10]. This narrative review aims to explore the advancements in lung ultrasound for neonatal RDS over the past decade, summarizing key findings and highlighting emerging debates in the literature on its advantages, limitations, and potential future applications in neonatal care.

## 2. Methodology

This narrative review was conducted to synthesize and evaluate the most relevant literature regarding the use of lung ultrasound (LUS) in diagnosing and managing neonatal respiratory distress syndrome (RDS) over the last 10 years. The following steps were followed to gather and analyze data:Search strategy: A comprehensive literature search was performed across multiple databases, including PubMed, Scopus, and Google Scholar. The search was restricted to peer-reviewed articles published between 2013 and 2023. Key search terms included “lung ultrasound”, “neonatal respiratory distress syndrome”, “LUS in neonates”, “RDS diagnosis”, and “ultrasound-guided surfactant therapy”;Inclusion and exclusion criteria: Studies were included if they focused on the use of lung ultrasound in neonates with RDS, provided data on diagnostic accuracy, reproducibility, or the use of LUS to guide treatment decisions. Only the studies published in English were included. Articles were excluded if they focused on adult populations, non-human studies, or did not specifically address RDS in neonates. Case reports and conference abstracts were also excluded due to a lack of rigorous peer review;Data extraction: Relevant data were extracted from each study, including study design, sample size, patient demographics, LUS techniques, diagnostic outcomes, and comparisons with other imaging modalities (e.g., CXR or CT). Information on LUS training and its impact on diagnostic accuracy was also collected;Analysis: The findings were summarized and analyzed to identify trends in the use of LUS for RDS diagnosis and management, as well as to assess the reproducibility and reliability of LUS scoring systems. Studies comparing LUS to traditional imaging methods like CXR were given particular attention, as were those exploring the role of LUS in guiding surfactant therapy and predicting the need for mechanical ventilation.

## 3. Lung Ultrasound Features in Neonates with Respiratory Distress Syndrome (RDS)

Lung scans obtained through trans-thoracic ultrasound in neonates are often limited by the presence of air within the lungs, which prevents the visualization of normal lung parenchyma [11,12,13,14,15,16]. Most of the images displayed on lung ultrasounds are artifacts rather than direct visualizations of the lung tissue. The superficial structures, including the skin, subcutaneous tissue, and thoracic muscles, form the outermost layers visible during the scan. The ribs, when scanned longitudinally, appear as curvilinear shapes with potential posterior acoustic shadowing, depending on whether the ribs are ossified or cartilaginous. A key anatomical structure visible in LUS is the pleural line, which appears as a thin, echogenic line, typically less than 0.5 mm wide. This line is consistently visible in healthy neonates just a few breaths after birth, and it corresponds to the interface between the parietal and visceral pleura [16].

One of the hallmark signs of a healthy lung is the “lung sliding” phenomenon, which reflects the movement of the pleurae (parietal and visceral) in sync with the respiratory cycle. The absence of lung sliding is a pathological finding that may indicate the presence of intrapleural air (pneumothorax) or fluid accumulation. Beyond the pleura-lung interface, the lung’s air content makes direct visualization difficult, but artifacts such as A-lines can be seen. A-lines are horizontal, equidistant, hyperechoic reverberation artifacts created by the pleural line’s reflection due to the acoustic impedance difference between the pleura and the aerated lung. These are normal in a healthy lung [16].

In cases of lung disease like RDS, the air content decreases and the lung density increases, leading to the appearance of vertical hyperechoic artifacts called B-lines. These B-lines originate from the pleural line and extend to the bottom of the ultrasound screen, moving with respiration. They indicate interstitial fluid accumulation [16]. However, it is important to note that a few non-confluent B-lines or comet-tail artifacts may also be observed in healthy neonates within the first 24 to 36 h after birth, as their lungs are transitioning from fluid-filled fetal lungs to air-filled postnatal lungs. Studies suggest that B-lines can persist for over two weeks in infants born at smaller gestational ages or those delivered via cesarean section [17,18,19,20].

As interstitial fluid increases and the severity of lung disease progresses, B-lines become more compact and numerous, ultimately resulting in a “white lung” appearance, which reflects significant pulmonary density. In severe cases, the ultrasound may show a lung image that closely resembles the appearance of a solid organ, indicative of a nearly complete absence of air in the lungs [16]. Several studies have confirmed the reliability of lung ultrasound in diagnosing RDS, with common findings such as pleural line irregularity, pleural thickening, and the presence of diffuse, compact B-lines [17,18,19].

In severe RDS, the “white lung” appearance—caused by pulmonary edema—is symmetrically distributed across both lung fields without spared areas, while less severe RDS is characterized by an interstitial syndrome with fewer B-lines [16,20]. Lung consolidation with air bronchograms is another important ultrasound feature of RDS. This feature helps distinguish RDS from other conditions such as neonatal transient tachypnea (TTN), which typically lacks lung consolidation. In mild RDS, lung consolidation may be small and subpleural with only a few air bronchograms, whereas in more severe cases, the consolidation can be extensive and air bronchograms more numerous [20]. Consolidations typically appear as irregular hypoechoic areas distinct from surrounding lung tissue and can be found in various lung fields, often bilaterally. Air bronchograms appear as dense, speckled, or snowflake-like structures [21,22].

Pleural effusion, either unilateral or bilateral, occurs in 15–20% of patients with RDS [23]. Chen et al. found that abnormal pleural lines, the absence of A-lines, and lung consolidation were present in all the RDS cases, while 88% of the cases exhibited a “white lung” or bilateral pulmonary edema [20]. The main ultrasound features that differentiate RDS from other neonatal respiratory conditions include lung consolidation with air bronchograms, pleural line abnormalities, pleural effusion, and compact B-lines [24,25].

A specific sign called the “lung pulse” has also been identified as an early and specific LUS marker of complete atelectasis, characterized by the pulsations synchronized with heartbeats in the absence of lung sliding. This sign has a sensitivity of 80% and a specificity of 100% for diagnosing RDS [6]. However, many of these findings are not exclusive to RDS, and the severity of the condition may cause variations in the ultrasound features seen in different areas of the lungs or even within the same lung [26]. A comprehensive diagnosis of RDS typically requires the concurrent presence of pleural line abnormalities, bilateral white lung, lung consolidation, and the absence of spared areas, in conjunction with clinical history and chest X-ray findings [16,27].

Several studies have demonstrated the diagnostic sensitivity and specificity of LUS in RDS. A 2022 study by Srinivasan et al. reported that lung consolidation with air or fluid bronchograms, white lungs, and the absence of spared areas achieved 100% sensitivity and specificity for diagnosing RDS [28]. In most cases, subpleural consolidations predominate, though in severe cases, more extensive consolidations may be observed. Pleural line abnormalities were present in all the neonates with RDS in the study, and the absence of A-lines and a white lung appearance had a sensitivity of 88% and a specificity of 94% [28]. Similarly, Liu et al. reported 100% sensitivity and specificity for the combination of lung consolidation, pleural line abnormalities, and bilateral white lung [6].

The evolution of LUS findings in neonates with RDS varies. As the condition improves, features such as less extensive lung consolidation, fewer air bronchograms, a reduction in pulmonary edema, normalization of the pleural line, and the reappearance of A-lines can be observed [20]. Conversely, LUS can also be useful for detecting RDS-related complications, including atelectasis, consolidation, and micro-abscesses [12].

Recently, new ultrasound markers, including ground-glass opacity signs (GOSs) and snowflake signs (SFSs), have been introduced for the diagnosis of RDS. Ground-glass opacity is often found in the early stages of RDS and can be confused with compact B-lines. Snowflake signs, which represent spots, patches, or thin-line air bronchograms within consolidated areas, suggest more advanced stages of RDS [26]. Snowflake patterns are considered highly specific for RDS and may serve as an important diagnostic marker in ultrasound imaging [26].

## 4. Lung Ultrasound Score in Neonates with Respiratory Distress Syndrome (RDS)

LUS has become an essential extension of the clinical examination for critically ill infants and children with respiratory distress, as recommended by the European Society of Pediatric and Neonatal Intensive Care (ESPNIC) guidelines [8,16,29]. Numerous studies highlight the benefits of LUS in newborns due to its ease of learning, bedside accessibility, reproducibility, and non-ionizing nature [8,16,29]. Given these advantages, LUS is increasingly considered the preferred initial bedside method for assessing infants with respiratory distress. However, reliance on traditional imaging modalities like CXR persists in some settings [30].

Beyond its diagnostic capabilities, LUS has evolved into a quantitative tool for assessing the severity of lung disease, making it a valuable “functional” technique [31,32]. Initial approaches relied on qualitative ultrasound pattern assessments, but quantitative scoring systems have since been developed to standardize and enhance the assessment of lung disease [32,33,34]. The lung ultrasound score (LUSco) is particularly useful for tracking disease progression and guiding therapeutic decisions by providing a dynamic evaluation of pulmonary aeration and quantifying disease severity [30,32].

To calculate a LUSco, the lungs are divided into zones, with the specific number of zones depending on the chosen scoring system [32,35]. A numerical value is then assigned to each zone based on the ultrasound findings, such as the presence of B-lines or consolidations. The overall LUSco is the sum of the individual zone scores, providing a comprehensive assessment of the severity and extent of lung disease, which also allows for objective comparisons between patients [35].

In neonates, who have smaller thoracic dimensions, a simplified scoring system is often used, with fewer pulmonary zones assessed [16]. Various scoring systems have been proposed, particularly to predict the need for surfactant replacement therapy in premature infants with RDS [30]. Typically, the chest is divided into anterior, lateral, and posterior regions, each of which is evaluated using standardized scanning protocols [32]. Each region is scored based on specific ultrasound findings, such as the presence of B-lines and subpleural consolidations [16].

One widely adopted LUSco was introduced by Brat et al. [24]. In their system, each lung is divided into three regions (upper anterior, lower anterior, and lateral). A score from 0 to 3 is assigned to each region: 0 for the presence of A-lines, 1 for three or more well-spaced B-lines, 2 for crowded and coalescent B-lines with or without subpleural consolidations, and 3 for extensive consolidations. The total LUS ranges from 0 (normal) to 18, reflecting the overall severity of lung disease [24].

Although Brat’s classification is widely used, other researchers have proposed alternative scoring systems with different pulmonary subdivisions. For example, the Rodriguez-Fanjul system assesses the anterior, lateral, and posterior zones of both hemithoraces [36], while the Raimondi system divides the lung fields along the hemiclavicular, anterior axillary, and posterior axillary lines in both hemithoraces [37]. Despite these differences, the scoring principles based on ultrasound findings remain consistent across systems [32].

In 2015, Brat et al. conducted the first major study to assess the predictive value of LUSco for determining the need for exogenous surfactant therapy in neonates with RDS [24]. This study demonstrated that LUSco could accurately predict the need for surfactant therapy based on early ultrasound findings before the onset of clinical symptoms. This enabled early-rescue surfactant therapy, potentially administered within the first two hours of life, even before clinical indicators necessitating treatment had fully developed [1]. The study included 130 neonates, with each lung divided into three regions (upper anterior, lower anterior, and lateral). A score of 0 to 3 was assigned to each region, and LUS was performed shortly after admission to the Neonatal Intensive Care Unit (NICU), before surfactant administration, following European guidelines. The researchers found significant correlations between LUSco and oxygenation indices, particularly in extremely premature infants. LUSCo was shown to reliably predict the need for surfactant therapy in preterm infants of a gestational age (GA) of under 34 weeks who were treated with CPAP from birth [32].

De Martino et al. also demonstrated the predictive accuracy of LUSco in determining the need for surfactant therapy in infants born at ≤30 weeks GA [14]. Their findings confirmed a significant correlation between LUS and oxygenation indices, underscoring the tool’s reliability, even when accounting for gestational age differences.

In a prospective study, Perri et al. compared the LUSco with CXR scoring systems to predict the surfactant administration in neonates with RDS. This study found that the LUSco had a higher Area Under the Curve (AUC) than the CXR scores in identifying the infants requiring surfactant [13]. Another study evaluated variations in the LUSco two hours after surfactant administration, revealing that a LUSco ≥ 7 predicted the need for a second surfactant dose with 94% sensitivity and 60% specificity [38,39].

A quality improvement initiative called the echography-guided surfactant therapy (ESTHER) project was conducted by Raschetti et al. in response to these findings [40]. The study found that the ESTHER method improved outcomes by increasing the number of neonates receiving surfactant within the first three hours of life, reducing peak FiO_2_ before surfactant administration, shortening the duration of invasive ventilation, and increasing ventilator-free days [32].

Most studies continue to utilize Brat’s scoring method for neonatal RDS, validating its effectiveness in guiding clinical management [24]. Overall, these studies indicate that the LUSco serves not only as an imaging tool but also as a functional instrument that facilitates early intervention and may outperform traditional examination methods in predicting the need for treatment. Table 1 shows a summary of the LUSco for the diagnosis of neonatal RDS.

Among the various LUS scoring systems, the Brat et al. scoring system has emerged as the most adaptable and widely validated approach, demonstrating high reproducibility across diverse clinical settings. Its straightforward methodology, focusing on key ultrasound findings such as A-lines, B-lines, and consolidations, ensures ease of use and reliability, even in resource-limited environments. While other systems like those proposed by Raimondi and Rodriguez-Fanjul provide comparable diagnostic accuracy, their complexity may limit widespread adoption. Therefore, the Brat system stands out as a practical and effective tool for standardizing LUS in neonatal respiratory distress syndrome, making it a strong candidate for broader clinical implementation and training.

While the LUS scoring systems offer significant diagnostic accuracy and reproducibility, real-world applicability is not without challenges. Operational barriers, such as variability in interpretation and the need for highly trained personnel, remain significant hurdles, particularly in resource-limited settings. Differences in operator expertise and familiarity with LUS can impact the consistency of scoring and diagnostic outcomes. Additionally, the learning curve for mastering LUS techniques and scoring systems may require dedicated training and ongoing validation efforts, which could limit their immediate adoption in some clinical environments. Addressing these challenges is essential to ensure that these systems can be effectively implemented in diverse healthcare settings.

## 5. Treating Respiratory Distress Syndrome (RDS)

Surfactant replacement therapy is the cornerstone of treatment for RDS, and remains the only pharmaceutical intervention with proven efficacy in managing this condition. As of 2024, the European Consensus Guidelines do not yet recommend LUS as a standard of care in treating RDS, nor do they advocate for the routine use of other radiological examinations. However, LUS is acknowledged as a valuable tool in aiding clinical decision-making and in reducing the need for CXR exposure by identifying specific signs of RDS [1,31].

The ability of LUS to predict clinical outcomes in RDS has been demonstrated by Raimondi et al., who showed that LU can predict CPAP failure in newborns with RDS with 100% specificity [41]. A major focus of recent research is to use LUS to predict the need for surfactant administration in preterm infants by assessing and quantifying the severity of lung disease. The echography-guided surfactant therapy (ESTHER) protocol, developed by Raschetti et al., has demonstrated that LUS-guided surfactant administration increases the effectiveness of surfactant treatment. It also reduces maximal FiO_2_ levels before surfactant is administered and shortens the duration of invasive ventilation [40].

However, some studies report conflicting findings. For instance, a recent study noted no significant changes in the total number of surfactant doses administered when implementing the ESTHER protocol, although LUS remains a powerful tool in identifying neonates at risk for CPAP failure. This allows clinicians to intervene earlier in the disease course, leading to more successful surfactant administration and a reduction in mortality, chronic lung disease, and air leaks in preterm newborns [30,42,43].

Corsini et al. compared three different LUS protocols developed by Brat et al. [24], Raimondi et al. [44], and Rodriguez-Fanjul et al. [36], which primarily differed in their lung partitioning and inclusion of posterior lung views. Despite these differences, all the protocols yielded excellent results in predicting the need for surfactant administration, provided a cut-off LUSco of 4 was used, which demonstrated 100% sensitivity [30]. Further studies have identified LUSco cut-off values of 7 and 10 as reliable indicators for administering the first and second doses of surfactant, respectively [45,46]. Additionally, LUSco can predict which preterm infants will not require a subsequent surfactant dose; a LUSco ≥ 7 calculated two hours after the first dose showed a 94% sensitivity and 60% specificity in identifying the neonates who did not need a second dose [38].

In a recent multicenter study, Raimondi et al. concluded that combining the LUSco with the FiO_2_ ratio significantly enhances the predictive power for determining surfactant need in preterm newborns, providing a more comprehensive and accurate approach to managing RDS [44]. Table 2 summarizes the key studies on lung ultrasound in predicting surfactant therapy and outcomes in neonatal RDS.

While LUS offers significant advantages in managing neonatal RDS, its application in certain contexts, such as high-frequency ventilation (HFV), presents notable limitations [47]. HFV introduces rapid, small-volume oscillations in the lungs that may complicate the interpretation of ultrasound artifacts, such as B-lines or pleural irregularities, potentially leading to the misinterpretation of lung aeration status. The dynamic lung changes during HFV may also mask subtle shifts in lung recruitment or overdistension that LUS typically detects in conventional ventilation. Additionally, the reliance on artifacts rather than the direct visualization of lung parenchyma underscores the inherent challenge of differentiating between atelectasis and interstitial fluid in rapidly ventilating neonates [48]. Operator dependency and the variability in skill levels further compound these issues, as interpreting subtle differences under HFV conditions requires advanced expertise. Standardized protocols for LUS during HFV are limited, making it difficult to integrate this tool effectively in such cases [49]. These limitations highlight the need for further research and protocol development to optimize the use of LUS in the context of HFV for neonatal RDS.

## 6. Further Uses of Lung Ultrasound in Neonatal Respiratory Pathology

Given the emerging role of LUS in neonatal care, numerous studies are ongoing to explore its broader applications in respiratory pathology. One prominent example is the LUNG study, which is investigating whether the early initiation of surfactant therapy—guided by LUS—can reduce the incidence of bronchopulmonary dysplasia (BPD) and mortality in preterm infants. This study builds on prior evidence suggesting that LUS may reduce the time to surfactant administration, potentially improving clinical outcomes [50].

As LUS becomes more widely adopted, new areas of research are being pursued to expand its utility. For instance, some studies are examining the measurement of pleural line thickness during different phases of the respiratory cycle in neonates with and without acute respiratory failure, which may offer new insights into lung mechanics and disease severity [51].

Additionally, LUS is being used to guide respiratory management beyond the immediate neonatal period. Research has shown that LUS can help predict the need for non-invasive respiratory support, such as nasal CPAP (nCPAP), and assess the risk of requiring mechanical ventilation [41,52,53]. By dynamically assessing lung function, LUS allows clinicians to make more informed decisions about respiratory support, particularly in the critical days following birth. Studies have shown that LUS is a pivotal tool in personalizing mechanical ventilation by enabling the real-time assessment of lung aeration and guiding the adjustment of positive end-expiratory pressure (PEEP) during invasive ventilation [54]. LUS visualizes lung dynamics, identifying areas of collapse and overdistension, which are critical in determining the optimal PEEP to achieve effective lung recruitment while minimizing ventilator-induced lung injury. Using LUS, clinicians can detect consolidation, atelectasis, or excessive aeration through specific patterns, such as B-lines or “white lung” appearances, which correlate with varying lung compliance and ventilation needs. Additionally, LUS-guided PEEP titration allows individualized adjustments to balance oxygenation improvement and prevents alveolar overdistension, thus supporting lung protective strategies and improving the outcomes for critically ill neonates and adults. Integrating LUS into ventilatory management minimizes the reliance on static imaging and enhances the precision in critical care [55].

LUS has also proven valuable in assessing lung recruitment during mechanical ventilation. Studies indicate that neonates whose lung recruitment is guided by ultrasound tend to experience faster reductions in FiO_2_ requirements, shorter hospital stays, and less lung inflammation. Moreover, these infants show reduced rates of barotrauma and spend less time on invasive ventilation, highlighting the potential of LUS for optimizing ventilatory strategies and improving long-term outcomes [56,57,58].

## 7. Future Perspectives on the Use of Lung Ultrasound in Neonates with Respiratory Pathology

LUS has proven to be a safe, reproducible, and accurate technique for bedside evaluation, especially when integrated with clinical data to improve diagnostic and therapeutic decisions. Its emerging role in neonatal care is particularly valuable in reducing the reliance on conventional radiology, thus minimizing unnecessary exposure to ionizing radiation in vulnerable infants. LUS is not intended to replace traditional imaging methods but rather to complement them, offering a dynamic, real-time tool for diagnosing and managing respiratory distress in neonates.

The development and validation of LUS scoring systems by various research groups have enhanced the clinical utility of LUS, aiding in the assessment of disease severity, guiding surfactant therapy, and predicting the need for mechanical ventilation in preterm infants. However, further research is needed to validate these scoring systems across different clinical settings, especially in low-resource environments where access to radiological tools is limited and operator experience may vary.

The growing body of literature over the past decade highlights LUS’ expanding applications beyond RDS, including its use in monitoring lung recruitment, guiding ventilation strategies, and assessing pleural line abnormalities. These applications show promise for optimizing respiratory management, shortening hospital stays, and reducing complications such as BPD and barotrauma. Despite these advances, one of the main limitations remains the variability in operator experience. This underscores the need for comprehensive, standardized training to ensure consistency and accuracy in NICUs and during specialist training programs. Providing structured training is essential to maximize LUS’ clinical potential and ensure consistent outcomes across different healthcare providers.

Combining LUS with biomarkers presents a promising approach to enhance the precision of protective mechanical ventilation strategies in neonatal RDS. Biomarkers, such as surfactant protein levels, inflammatory markers (e.g., interleukins or cytokines), and indicators of alveolar epithelial injury, can provide biochemical insights into the state of lung injury and repair processes. When used alongside LUS, biomarkers can help validate findings and offer a more comprehensive understanding of lung pathology. For instance, elevated inflammatory markers could corroborate ultrasound-detected patterns of interstitial fluid accumulation or consolidation, signaling the need for cautious PEEP adjustments to prevent the overdistension or exacerbation of injury [59]. Additionally, monitoring biomarker trends during LUS-guided interventions may provide an early indication of response to ventilation strategies, such as the efficacy of lung recruitment maneuvers or surfactant administration [60]. This integrated approach could help tailor ventilation parameters to individual needs, ensuring optimal lung protection while minimizing ventilator-induced lung injury. However, the logistical challenges of timely biomarker analysis and the need for further validation in clinical practice remain areas for future research. LUS has significant potential to be integrated into routine clinical protocols for neonates, even across diverse healthcare settings, including low-resource environments [61,62]. Its portability, non-invasive nature, and capacity for real-time diagnostic imaging make it particularly advantageous where traditional modalities like X-rays or CT scans are unavailable or impractical. To enhance its applicability, simplified protocols and user-friendly scoring systems, such as the Brat et al. method, should be prioritized. These systems are designed to require minimal training, facilitating easier adoption across varying levels of expertise. Additionally, leveraging advancements like pre-programmed scoring algorithms and artificial intelligence-assisted interpretation can help mitigate operator dependency by providing standardized, automated guidance, enabling less experienced clinicians to use LUS effectively. Coupled with targeted training initiatives and telemedicine for remote expert support, these innovations could democratize LUS usage, transforming it into a cornerstone of neonatal respiratory care in low-resource environments. However, practical challenges remain significant and cannot be overlooked. Implementing LUS widely requires addressing the need for trained personnel, a notable barrier in settings with limited opportunities for specialization and overburdened healthcare staff. Operator-dependent variability in interpretation further complicates consistent use and may impact diagnostic accuracy. Additionally, the initial costs of acquiring and maintaining portable ultrasound devices, along with the absence of standardized training programs tailored to resource-constrained environments, present formidable obstacles. While AI-assisted solutions offer promise, they are still in the early stages of validation and may not be readily accessible in low-resource settings. Overcoming these challenges demands a multifaceted approach that includes developing cost-effective equipment, scalable training modules, and simplified protocols tailored for these environments. Without such efforts, the transformative potential of LUS to enhance neonatal care and improve outcomes globally may remain unrealized. Prioritizing research and investment in scalable, equitable solutions is essential to unlock the full potential of LUS and ensure its broader adoption.

Moreover, while LUS has demonstrated significant promise in improving diagnostic accuracy and guiding interventions for neonatal RDS, there remains limited large-scale evidence directly linking LUS-guided interventions to substantial improvements in long-term outcomes, such as reductions in BPD or mortality. Most studies to date have focused on short-term outcomes, such as predicting surfactant need or CPAP failure, rather than evaluating comprehensive, long-term clinical impacts. The assumption of a direct correlation between LUS use and improved outcomes like reduced BPD or mortality is therefore not yet fully substantiated by existing data. Future research should focus on large-scale, multicenter studies that assess the long-term benefits of LUS-guided protocols to provide more definitive evidence supporting these conclusions. This perspective ensures a balanced understanding of LUS’ current role and its potential for broader clinical impact.

The limitations of this review stem from several factors. First, its narrative nature does not allow for a formal meta-analysis due to the heterogeneity of the study designs, the LUS protocols, and the scoring systems reviewed. This variability may limit the generalizability and consistency of the findings. Second, the inclusion of studies is restricted to English-language publications, which could introduce selection bias and exclude potentially relevant research. Third, operator dependency and variability in expertise in conducting and interpreting LUS are significant challenges that could affect the reproducibility of the results. Additionally, most studies rely on small sample sizes or single-center experiences, which may not be representative of broader populations. Lastly, the review does not extensively address the limitations of LUS in specific contexts, such as during high-frequency ventilation, where the interpretation of artifacts can be particularly challenging. These factors underscore the need for more standardized training, multicenter validation studies, and the development of uniform LUS protocols for neonatal care.

## 8. Conclusions

LUS has proven to be a safe, reproducible, and accurate technique for bedside evaluation, especially when integrated with clinical data to improve diagnostic and therapeutic decisions. Its emerging role in neonatal care is particularly valuable in reducing the reliance on conventional radiology, thus minimizing unnecessary exposure to ionizing radiation in vulnerable infants. LUS is not intended to replace traditional imaging methods but rather to complement them, offering a dynamic, real-time tool for diagnosing and managing respiratory distress in neonates. Future research should prioritize the development of standardized training protocols to address the variability in operator expertise. In addition, it is important to validate LUSco in diverse clinical settings, particularly those with limited resources. Investigating the long-term impact of LUS-guided interventions on neonatal outcomes, such as the reduction in BPD and mortality, will also be critical. Finally, exploring technological advancements such as the integration of artificial intelligence (AI) for automated interpretation could further enhance the accuracy and usability of LUS, especially for less experienced operators. By addressing these priorities, LUS can become an even more integral tool in neonatal respiratory care, improving both short- and long-term outcomes for newborns with respiratory distress.

## Figures and Tables

**Table 1 diagnostics-14-02793-t001:** Summary of lung ultrasound scoring systems for neonatal respiratory distress syndrome (RDS).

Scoring System	Lung Regions Assessed	Scoring Criteria	Total Score Range	Key Features
Brat et al. [24]	6 regions: upper anterior, lower anterior, lateral (both lungs)	0: A-lines; 1: ≥3 B-lines; 2: Crowded/coalescent B-lines with/without consolidation; 3: Large consolidations	0–18	First widely used scoring system for neonatal RDS
Raimondi et al. [37]	6 regions: anterior, lateral, posterior (both lungs)	0: A-lines; 1: ≤2 B-lines; 2: Numerous B-lines or consolidations; 3: Extensive consolidations	0–18	Includes posterior lung fields in the scoring system
Rodriguez-Fanjul et al. [36]	6 regions: anterior, lateral, posterior (both lungs)	Similarly to Brat et al., focusing on lung consolidation and B-lines	0–18	High accuracy in predicting CPAP failure

**Table 2 diagnostics-14-02793-t002:** Key studies on lung ultrasound in predicting surfactant therapy and outcomes in neonatal respiratory distress syndrome (RDS).

Study	Study Population	Key Findings	Clinical Implications
Brat et al. [24]	130 preterm neonates < 34 weeks GA	LU accurately predicted the need for surfactant therapy based on early ultrasound findings, before clinical symptoms	Enables early surfactant therapy (“early-rescue”), potentially improving outcomes
De Martino et al. [14]	Preterm neonates ≤ 30 weeks GA	Significant correlation between LU score and oxygenation index, with LU predicting surfactant need	Reinforces LU as a reliable tool for predicting surfactant therapy in very preterm infants
Perri et al. [38]	Preterm neonates with RDS	LU had a higher AUC than CXR scores in predicting the need for surfactant treatment	Suggests LU may outperform CXR in identifying neonates requiring surfactant
Raschetti et al. [40]	Preterm neonates in NICU	Echography-guided surfactant therapy (ESTHER) resulted in earlier surfactant administration and reduced FiO_2_ levels	Demonstrates the benefits of LU-guided therapy in reducing invasive ventilation and improving clinical outcomes
Raimondi et al. [44]	Multicenter study on preterm infants	Combination of LU score and FiO_2_ ratio had the highest predictive power for determining surfactant need	Suggests that combining LU with clinical parameters enhances decision-making for surfactant therapy

## Data Availability

Not applicable.

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
