# Peer review of "Lung Ultrasound in Neonatal Respiratory Distress Syndrome: A Narrative Review of the Last 10 Years"

_diagnostics, 2024, doi:10.3390/diagnostics14242793_

Round 1
Reviewer 1 Report
Comments and Suggestions for Authors
In this narrative review, the authors describe the use of US in the context of ARDS. Overall, the paper is interesting and offers an apprapriate overview on the use of lung US in neonatal ARDS in the last decade. The analysis of the current literature is updated and the discussion is well balanced. I have some minor comments for the authors in order to improve the quality of the paper:
- Firs of all, please explaine the role of Lung ultrasound to personalize the mechanical ventilation and how can be used to identify an appropriate PEEP level during invasive ventilation.
- I think that the authors should underline better the potential limitations of the use of the ultrasound in the context of neonatal ARDS in particular regarding the use of High frequency ventilation.
- Please take into account the potential role of biomarkers in combination of Lung ultrasound to guarantee a protective mechanical ventilation.
in conclusion, i think that this narrative review is of particular interest for the scientific audiance and could be offer an interesting perspective for further investigations
- i suggest to summerize the main indications of lung ultrasound in a figure that could show how it is possible to use the US in this specific context.
-
Author Response
In this narrative review, the authors describe the use of US in the context of ARDS. Overall, the paper is interesting and offers an appropriate overview on the use of lung US in neonatal ARDS in the last decade. The analysis of the current literature is updated and the discussion is well balanced.
Re: Thank you for your positive evaluation. We improved the manuscript according to your suggestions.
I have some minor comments for the authors in order to improve the quality of the paper:
- Firsi of all, please explaine the role of Lung ultrasound to personalize the mechanical ventilation and how can be used to identify an appropriate PEEP level during invasive ventilation.
Re: Thank you for your suggestion. We added a paragraph on this topic with two new references (p. 9).
- I think that the authors should underline better the potential limitations of the use of the ultrasound in the context of neonatal ARDS in particular regarding the use of High frequency ventilation.
Re: Added as requested (p. 8).
- Please take into account the potential role of biomarkers in combination of Lung ultrasound to guarantee a protective mechanical ventilation.
Re: Added a paragraph on this topic (p. 10).
in conclusion, i think that this narrative review is of particular interest for the scientific audiance and could be offer an interesting perspective for further investigations.
Re: Thank you very much for your comments. We revised the manuscript accordingly.
- i suggest to summerize the main indications of lung ultrasound in a figure that could show how it is possible to use the US in this specific context.
Re: We think that the two Tables are appropriate for the purpose of the review and a Figure could not add useful information for the readers.
Reviewer 2 Report
Comments and Suggestions for Authors
The authors of this narrative review aimed to explore the advancements in lung ultrasound for neonatal Respiratory Distress Syndrome over the past decade, summarizing key findings and highlighting emerging debates in the literature on its advantages, limitations, and potential future applications in neonatal care.
Methodology is well presented, clear and concise. However, I recommend that you move limitations from methodology into the discussion part.
The two tables are clear and correct.
I consider that the conclusions are a bit long, they should be shorter and clearer.
The article presents 64 references, being up to date.
Being a narrative review, I recommend you have at least 70 references, this way you will expand the article.
Author Response
The authors of this narrative review aimed to explore the advancements in lung ultrasound for neonatal Respiratory Distress Syndrome over the past decade, summarizing key findings and highlighting emerging debates in the literature on its advantages, limitations, and potential future applications in neonatal care.
Re: Thank you for your positive evaluation. We improved the manuscript according to your suggestions.
Methodology is well presented, clear and concise. However, I recommend that you move limitations from methodology into the discussion part.
Re: Revised as suggested (p. 10).
The two tables are clear and correct.
Re: Thank you for the appreciation of our Tables.
I consider that the conclusions are a bit long, they should be shorter and clearer.
Re: We divided the text in two paragraphs because the other reviewers asked us to add further comments. In this way, the Conclusions are shorter and cleared (pp. 10-11).
The article presents 64 references, being up to date. Being a narrative review, I recommend you have at least 70 references, this way you will expand the article.
Re: We added further references and now the overall number is higher than 70.
Reviewer 3 Report
Comments and Suggestions for Authors
File attached

nice
Author Response
Re: Thank you for your comments. We revised our manuscript according to your recommendations.
- A comparison between LUS and other diagnostic modalities (like chest X-rays or CT scans)
in terms of specificity, sensitivity, or overall effectiveness should have been mentioned in
the abstract.
Re: Added as suggested (p. 1).
- While the abstract emphasizes the advantages and potential of LUS, it only briefly mentions
challenges in standardizing training and validating its use. LUS may not be available or
feasible in all clinical settings, especially in resource-limited environments.
Re: Clarified as suggested (p. 1).
- Why the exclusion of case reports and conference abstracts. While these types of studies
may lack the rigorous peer review of full-length articles, they can still provide valuable
insights, especially in emerging fields like lung ultrasound (LUS) in neonates.
Re: We excluded case reports and conference abstracts because, while they can provide valuable preliminary insights, they often lack the methodological rigor and peer-review processes required to ensure reliability and reproducibility. The field LUS in neonates has over a decade of well-established literature, allowing us to focus on robust studies that provide validated evidence and generalizable conclusions. Including less rigorous sources could dilute the quality of our synthesis and compromise the reliability of the findings presented. Thank you for your understanding.
- clarifying if any scoring system is more adaptable or has demonstrated superior
reproducibility across diverse clinical settings would strengthen the argument for one
approach over others
Re: Clarified (p. 7).
- operational challenges in adopting these scoring systems, such as variability in
interpretation or the need for highly trained personnel, could provide a more balanced
view of their real-world applicability.
Re: Clarified (p. 7).
- how do you foresee the potential for LUS to be integrated into routine clinical protocols for
neonates across diverse healthcare settings, particularly in low-resource environments
where access to highly skilled operators and advanced technology may be limited?
Re: Clarified (p. 11).
- While LUS has shown promise, there is still limited large-scale evidence directly linking
LUS-guided interventions to substantial improvements in long-term outcomes like
reducing bronchopulmonary dysplasia (BPD) or mortality. This conclusion assumes a
direct correlation that may not yet be fully supported by existing data.
Re: Added (p. 11).
- The conclusion acknowledges variability in operator experience but does not fully address
the practical challenges of implementing LUS widely, especially in low-resource settings.
these issues are easily surmountable, which might not be the case.
Re: Added (p. 11).
Round 2
Reviewer 3 Report
Comments and Suggestions for Authors
now the content has improved a lot.